# Allometric Characteristics of Rice Seedlings under Different Transplanted Hills and Row Spacing: Impacts on Nitrogen Use Efficiency and Yield

**DOI:** 10.3390/plants11192508

**Published:** 2022-09-26

**Authors:** Xiaoyan Wu, Izhar Ali, Anas Iqbal, Saif Ullah, Pengli Yuan, Anjie Xu, Dongjie Xie, Yuxi Zhou, Xinlu Long, Hua Zhang, Jing Yu, Zixiong Guo, He Liang, Shanqing Wei, Ligeng Jiang

**Affiliations:** 1Key Laboratory of Crop Cultivation and Farming System, College of Agriculture, Guangxi University, Nanning 530004, China; 2Guangxi Key Laboratory of Agro-Environment and Agro-Products Safety, College of Agriculture, Guangxi University, Nanning 530004, China

**Keywords:** rice, nitrogen accumulation, nitrogen use efficiency, allometric growth, number of seedlings per hill, row spacing, yield

## Abstract

The number of seedlings per hill and the configuration of plant row spacing are important management measures to improve rice yield. In the present study, we evaluated the impact of various seedlings per hill (1, 3, 6, and 9 seedlings hill^−1^) under four different rice verities (two conventional rice, two hybrid rice) on allometric characteristics, nitrogen use efficiency (NUE) and yield in 2020 at early and late season. Results showed that compared with nine seedlings per hill (wide row spacing), the number of effective panicles, yield, grain biomass allocation, grain-to-leaf ratio, grain nitrogen accumulation, nitrogen dry matter production efficiency (NDMPE), N harvest index (NHI) of 1 seedling per hill increased by 21.8%, 10.91%, 10.5%, 32.25%, 17.03%, 9.67%, 6.5%, respectively. With the increase of seedlings per hill and the expansion of row spacing, stem biomass (SB) and reproductive biomass (RB) increased with the increase of above-ground biomass, mainly showing the relationship of isometric growth. Leaf biomass (LB) increased with above-ground biomass, mainly showing the relationship of allometric growth. The results suggested that under the same basic seedlings, transplanting 1 seedling per hill and dense planting was the most beneficial to improve rice yield.

## 1. Introduction

Rice (*Oryza sativa* L.) is a staple food for more than half of the world’s population, and about 90% of rice is produced and consumed in Asia [1]. Due to the shortage of arable land and water resources, it is not possible to increase the rice production area [2]. Therefore, enhancing yield per unit area is always an important goal of crop cultivation and breeding, and it is also an important direction of rice research in China [3,4]. Furthermore, planting densities and nutrient management are important agronomic parameters that must be optimized to achieve high grain yields. Rice is a monocotyledonous plant [5], and tillering is one of its characteristics. An excessive number of seedlings on a single hill and suboptimal plant row spacing will reduce the effectiveness of rice tillering [6]. The main stems and tillers of rice compete for resources such as light, air, and nutrients [7]. Therefore, an appropriate planting density can effectively increase yield, while improper planting density can decrease grain yields. Transplanting density and ecological site both have significant effects on rice yield, source-sink relationship, and dry matter accumulation [8]. Therefore, to establish a high-efficiency rice population structure, the number of rice seedlings in each hill and plant spacing must be strictly managed to maximize the resource and light utilization efficiency of the rice population [9].

Competition is prevalent in the agroecosystem [10]. Within a rice population, competition is primarily intraspecific, and the competition between individual plants is essentially the competition at the organ level for limited resources [11]. Competition causes changes in trait expression in plants [12]. Seeding density is a major factor affecting plant plasticity as it determines the intensity of intraspecific competition for interactions between plants in different environments [13]. In the process of rice development, the result of rice resource competition will ultimately be reflected in yield [14]. Competitiveness is closely related to functional traits such as biomass allocation. Therefore, it is necessary to quantify the apparent and true plasticity of functional traits [15]. Every plant expresses a certain allometric growth pattern under specific conditions; that is, plants exhibit allometric plasticity, which changes quantitative relationships between the growth and distribution of organs in an individual. The allometric growth of the plant itself is the “apparent plasticity,” while the change of the allometric growth curve is the true plasticity [16]. Allometric characteristics of individual rice can be obtained by allometric analysis.

Planting structure not only affects the plasticity of rice phenotype but also affects nitrogen use efficiency. Nitrogen is a macronutrient for plant growth and development, and the increase in inorganic nitrogen usage is vital to improving crop growth and yield. Studies have shown that increasing planting density can compensate for the adverse effects of reduced nitrogen application and improve nitrogen use efficiency [17]. In the early stage of plant growth, leaves are the nitrogen storage reservoir, and nitrogen is redistributed into the grain. The source-sink relationship of nitrogen exists among the roots and leaves in the early stage of plant growth and among the leaves and seed development in the mature stage [18]. In cereal crops, 50–90% of the nitrogen in the grain is mainly transported through leaf nitrogen [19]. Factors such as nitrogen distribution among stems and leaves, photosynthetic efficiency, rubisco activity, and leaf senescence determine nitrogen use efficiency [20]. For instance, San-oh et al. found that the number of seedlings planted per hill affects rice’s photosynthetic rate and physiological process. Seedlings per hill affect the Ribulose bisphosphate carboxylase oxygenase/oxygen Enzyme (Rubisco) and nitrogen levels; there is a positive correlation between Rubisco levels and photosynthetic rate and between nitrogen levels, and Rubisco, the levels of Rubisco and N in the leaves of rice planted with one seeding per hill are higher; thus the leaf photosynthetic rate is higher during the maturation process [21].

In this experiment, under the same basic seedling number, the allometric growth characteristics of rice under the synergistic change of rice seedlings per hill and plant row spacing were investigated, and the effect on yield and nitrogen use efficiency, which was significant for improving the efficiency of rice planting and simplifying the production process. Therefore, the objectives of the study were: (1) To explore the effect of synergistic changes in the number of seedlings per hill and row spacing on rice yield under the same basic seedlings; (2) To study the effect of synergistic changes in the number of seedlings per hill and row spacing on the biomass distribution and allometric growth characteristics of above-ground organs; (3) To compare the differences in nitrogen use efficiency of rice under different seedlings per hill and row spacing.

## 2. Results

### 2.1. Yield and Yield Components

Rice yield was significantly affected by variety and treatment (number of seedlings per hill × row spacing). Still, the interaction between variety and treatment had no significant (*p* > 0.05) effect on yield (Table 1). Compared with V1, V2, and V3 varieties, the average yield of V4 was increased by 23.99%, 19.84%, and 19.62% in the early season and 21.02%, 8.14%, and 8.73% in the late season, respectively. Among the four treatments, the yield decreased gradually with the increase of seedlings per hill (the spacing between plants and rows gradually increased). The yield of T1 in the early season increased by 5.15%, 6.40%, and 16.27% compared with that of T2, T3, and T4, respectively. While in the late season, the yield of T1 increased by 8.31%, 19.65%, and 27.32% compared with that of T2, T3, and T4, respectively.

The analysis of the yield components of the four rice varieties in the early and late seasons showed that with the increase in the number of seedlings per hill and row spacing (Table 2), the number of effective panicles and spikelets per panicle showed significant differences (*p* < 0.05) among different treatments. In the early and late seasons, compared with T4, the number of effective panicles of T1 increased by 11.26% and 10.55%, respectively. Spikelets per panicle only showed the difference in the late season; T1 increased by 13.17%, 13.43%, and 11.93% than T2, T3, and T4. From the correlation heat map of yield and yield component factors (Figure 1), it can be seen that yield is significantly positively correlated with the number of effective panicles and the percentage of filled grains.

### 2.2. Biomass Allocation, Ratio of Grain to Leaf

Grain biomass accumulation is of great significance to the final yield of rice. The results of above-ground biomass distribution of rice showed that biomass allocation was significantly (*p* < 0.05) affected by variety, number of seedlings per hill× row spacing (Table 3), and the organ biomass allocation of rice from large to small was as follows: grains > stems > leaves (Figure 2A,B). Among the four varieties (Figure 2C), the grain biomass allocation of V4 was significantly higher than that of V1, V2, and V3 by 12.7%, 5.1%, and 6.3%, respectively, in the early season. Whereas, in the late season, the grain biomass allocation of V4 was not significantly different from V2 and V3. Among the four treatments, the grain biomass allocation proportions of T1 treatment of two hybrid rice (V3, V4) were increased by 63.22% and 65.82%, respectively, in the early season, which was significantly higher than those of T2, T3, and T4. Similarly, in the late season, except for V3, the grain biomass allocation of V1, V2, and V4 in the T1 treatment was significantly higher than that in T2, T3, and T4 treatment, and the ratio was between 56.7% and 61.0%. Compared with T4 (Figure 2D), in the early season, the average grain allocation of the T1 treatment increased by 7.12%, 9.40%, and 6.55%, respectively. In the late season, the average grain allocation of the T1 treatment increased by 10.30%, 12.58%, and 14.45%, respectively. The proportion of stem biomass allocation decreased with increasing grain biomass allocation.

The grain-to-leaf biomass ratio reflects the plant’s strategy for allocating resources between reproductive and production organs, with higher grain-to-leaf ratios leading to higher yield potential. The grain-to-leaf ratio was significantly affected by variety and treatment (Figure 3). Among the four varieties, V4 in the early season had the highest grain-to-leaf ratio (7.34). The grain-to-leaf ratio of V2 in the late season was the highest (7.42). Among the four treatments, compared with the T4 treatment, the T1 treatment had the largest grain leaves (no significant difference in the early season but a significant difference in the late season). The average grain leaves the ratio of T1 increased by 26.81% and 37.68% in the early and late growing seasons, respectively, compared with T4 treatment.

### 2.3. Allometric Growth between Above-Ground Organs

Allometric analysis of the above-ground organ biomass showed that stem, leaf, and grain biomass increased with the increase in individual plant biomass (Table 4, Figure 4A and Figure 5A). When the relative rates of growth are not equal, allometric and isokinetic growth differentiation occurs. Table 4 shows that the stem biomass and biomass per plant showed six isokinetic growth relationships and two allometric growth relationships were present in early and late season rice; leaf biomass and per plant biomass showed three isokinetic and two allometric growth relationships in early and late season rice. Five allometric growth relationships, seven isokinetic growth relationships, and one allometric growth relationship in both reproductive biomass and biomass per plant were present in early and late-season rice. This indicated that allometric growth patterns were prevalent between leaf biomass and single plant biomass, while isokinetic growth patterns prevailed between stem biomass, grain biomass, and single plant biomass. Among treatments with different numbers of seedlings, T1 and T4 exhibited three isokinetic and three allometric growth relationships in early and late season rice. In comparison, T2 and T3 showed only 1 and 2 allometric growth relationships, respectively.

Table 5, Figure 4B and Figure 5B show that there is a linear relationship between grain organs and stems, while the linear relationship between leaves and stems is not obvious. Table 5 shows 6 allometric and 18 isokinetic growth relationships in the early and late season rice, and the isokinetic growth relationship was dominant among the organs. Among the six allometric growth relationships, GB-LB appeared four times. Among treatments with different numbers of seedlings, T1, T2, T3, and T4 showed 3, 2, 0, and 1 allometric growth and 3, 4, 6, and 5 isokinetic growth relationships in the early and late seasons, respectively.

The common slope test and drift type analysis were carried out on the allometric relationships between rice organ biomass and different seedlings. The growth relationships were all D-type (Table 6).

### 2.4. Nitrogen Accumulation and Nitrogen Use Efficiency

Nitrogen accumulation in above-ground organs of rice showed the trend in descending order: grains > stem > leaf (Table 7). Rice varieties had significant effects on nitrogen accumulation in stems (SN), nitrogen accumulation in leaves (LN), and nitrogen accumulation in grains (GN). In the early season, the varieties with the highest SN were V1 and V3, while the LN of V1, V2, and V3 also remained at a high level. V3 has the highest GN (223.78 kg kg^−1^) and the final TNA (316.01 kg kg^−1^). In the late season, the SN and LN of V1 and V4 were higher than those of V2 and V3, and the GN and TNA of V4 were the highest, which were 184.19 kg kg^−1^ and 262.46 kg kg^−1^, respectively. The number of seedlings per hill × row spacing also significantly affected grain nitrogen accumulation (except SN in the early season). Nitrogen use efficiency (NUE) is an important index for evaluating rice growth status. The number of seedlings per hill × row spacing significantly affected N dry matter production efficiency (NDMPE) and N harvest index (NHI), and the NDMPE of V1 and V3 gradually decreased with the increase of the number of seedlings per hole and row spacing, in the early season (Table 8). From the average value of the four treatments, the NDMPE of T1 was significantly increased by 8.58%, 11.40%, and 12.39% than that of T2, T3, and T4. Compared with T2, T3, and T4, the NHI leaves of T1 increased significantly by 3.22%, 8.26%, and 5.16%. The differences in NGPE among the four treatments were not significant. In the late season, the number of seedlings per hill × row spacing significantly affected NDMPE, NGPE, and NHI. The NDMPE, NGPE, and NHI of V1 decreased with the increase in the number of seedlings per hill and row spacing.

From the mean of the four treatments (Table 9), it can be found that in the early season, the SN and LN of T3 were the largest among the four treatments, which were 66.34 kg hm^−2^ and 30.96 kg hm^−2^, respectively. In the late season, the SN and LN of T2 were the largest, which were 51.11 kg hm^−2^ and 25.05 kg hm^−2^, respectively. The GN of the T1 treatment was the highest in both growing seasons, with a significant increase of 14.06% (early season) and 20.0% (late season) compared with the T4 treatment. From the average of the four varieties (Table 9), the NDMPE of T1 was significantly increased by 4.32%, 4.81%, and 6.40% than that of T2, T3, and T4. The NGPE of T1 was significantly increased by 11.95%, 12.89%, and 17.09% than that of T2, T3, and T4. The NHI of T1 was significantly increased by 7.95%, 7.83%, and 7.83 than that of T2, T3, and T4. In addition, among the four varieties, the NDMPE of V1 was the highest in both growing seasons, the varieties with the highest NGPE were V4 and V2, and the varieties with the highest NHI were V4 and V3.

## 3. Discussion

In this study, ‘Yexiangyou2’ (V4) was a hybrid rice with strong tillering ability, with the highest yield, grain biomass allocation (early season), grain-to-leaf ratio (early season), NGPE, and NHI. The largest variety with NDMPE was ‘Guiyu9’ (V1), ‘Zhenguiai’ (V2) also had high NGPE, and ‘Zhuangxiangyou5’ (V3) had the highest GN, TNA, and NHI. Numerous studies have been carried out to investigate the effects of seedlings per hill and row spacing on rice growth [22,23]. Transplanting 1–2 seedlings per hill has a wide range of sources of rice tillers. Transplanting 3–4 seedlings per hill can promote the occurrence of the central tillering position, which is the main source of tiller panicles and total panicles [24]. However, the study by Wiangsamut et al. [25] showed that transplanting 1 seedling per hill yielded higher yields than 4 seedlings per hill saving production costs. At the same time, there are many discoveries in the study of densely planted rice. At high planting density (25 cm × 11 cm), although the number of panicles per square meter of single seedling machine transplanting (SMT) was lower than that of conventional machine transplanting (CMT), the number of spikelets per panicle and the grain filling rate of each panicle were higher [26]. Machine-planting single seedlings at high density can improve single tiller performance by reducing non-productive tillers, increasing bank size by increasing secondary shoots per panicle, and increasing dry matter yield in late heading, thereby increasing grain yield [27]. In this experiment, yield was significantly affected by the number of effective panicle and percentage of filled grains. The number of effective panicles of T1 was higher than that of T4; in the two growing seasons, the yield of 1 seedling hill^−1^ (T1) increased by 16.27% and 27.32%, respectively, compared with 9 seedlings hill^−1^ (T4), with an average increase of 21.80%.

The regulation of biomass allocation among plant organs is a survival strategy for plants to cope with environmental changes. It has been shown that the allocation of organ biomass is specific among species and influenced by plant size and growing environment [28]. Restrictive regulation of population density and individual growth is the key to determining biomass allocation in competitive growth environments [29,30,31]. When the number of seedlings transplanted in each hill is increased in rice planting, the competition for resources among individuals intensifies [32]. This intensified competition between plants on a hill will inevitably lead to changes in biomass allocation strategies, and the result of our study showed that the row spacing of T1 (12.93 cm) was narrower than that of T4 (38.80 cm). Under the conditions of this experiment, the row spacing of T1 (12.93 cm) is narrower than that of T4 (38.80 cm), but T1 is more conducive to increasing the biomass distribution of grains than T4. In the early and late seasons, the distribution of grain biomass in T1 treatment ranged from 56.7% to 65.82%, and the ratio of grain to leaf increased by 26.81% and 37.68% compared with T4.

The result of differences in biomass allocation between organs is a change in allometric relationships. This study showed that the allometric relationship appeared more frequently in T4, and the allometric relationship between leaf biomass and aboveground biomass was the most obvious. Among the six allometric relationships of leaves, stems, and grain organs, four were related to leaves. In addition, leaf and grain organs showed c-type allometric growth, indicating that the differences in allometric characteristics of T1 and T4 were related to the significant differences in individual biomass. The leaves of the plant will change accordingly as the density changes [33]. With the increase in altitude, the leaf length, leaf width, girth, and other functional traits of the leaves bamboo decreased significantly, but the leaves of the middle-altitude bamboo species had the highest specific leaf area and the lowest leaf dry matter content, and the change of altitude was obvious. Inspired bamboo growth potential and morphological plasticity [34]. When testing the response of rape (*Brassica napus* L.) to temperature, it was also found that the leaf mass per area (LMA) at high temperature was significantly smaller than that at medium and low temperature, but the leaves at high temperature were significantly wider, and the leaves grew through the leaves. Modeling the functional structure of plants can predict plant growth under different environmental conditions [35]. Allometric growth allows plants to adapt to environmental changes and maximize favorable phenotypes to increase their niche breadth [36,37,38]. This also proves that the regulatory function of leaves plays an important role in plant growth when the environment changes.

In addition, there were also differences in nitrogen accumulation in stem, leaf, and panicle organs. Among the four varieties, the nitrogen accumulation in the grain was the highest. Compared with the T4 treatment (4 seedlings per hill, row spacing 38.80 × 38.80 cm), the T1 treatment (1 seedling per hill, row spacing 12.93 × 12.93 cm) had higher N accumulation, dry matter N use efficiency, and N harvest index, indicating that 1 seedling per hill and narrow rows were more conducive to improving N use efficiency. Because of the low efficiency of nitrogen use, only a very minor part of the nitrogen applied to the soil is absorbed by the rice seedlings. In contrast, the excess nitrogen applied can have a negative impact on the environment [39]. To improve nitrogen use efficiency, researchers studied a reduced nitrogen densely planting (RNDP) cultivation mode, which can make rice obtain similar yields to conventional high-yield practice (CHYP). This is due to the increased storage capacity of grains per panicle per unit area, increased biomass accumulation after heading, and improved nitrogen use efficiency [17]. It can be seen that dense planting plays a crucial role in improving the rice yield. When studying the interaction effect of nitrogen application rate and density on rice grain yield and nitrogen use efficiency, Hou et al. [40] found that nitrogen recovery efficiency (NRE) increased by 12.7 to 40.0% with the increase of planting density.

## 4. Materials and Methods

### 4.1. Experiment Site, Time, and Materials

The experiment was carried out in an experimental field at the Rice Research Institute of Guangxi University (N22°50′28.41″ N, E108°17′9.00″ E) in the early (April to July) and late seasons (August to November) of 2020. A total of four rice cultivars were tested, including two conventional rice varieties (‘Guiyu9’and ‘Zhenguiai’) and two hybrid rice varieties (‘Zhuangxiang5’ and ‘Yexiangyou2’). In this article, V1, V2, V3, and V4 are used to represent ‘Guiyu9’, ‘Zhenguiai’, ‘Zhuangxiang5’, and ‘Yexiangyou2’. The “Zhenguiai” seeds were obtained from the research group, which maintains this variety, and the seeds of the other varieties were provided by the breeding unit of Guangxi University. Before the experiment, the basic properties of the soil were: pH 6.7, total nitrogen 1.72 g kg^−1^, total phosphorus 1.62 g kg^−1^, total potassium 5.90 g kg^−1^, hydrolyzed nitrogen 187.4 mg kg^−1^, organic carbon 18.3 g kg^−1^, and organic matter 31.48 g kg^−1^.

### 4.2. Experiment Design

The experiment was laid out in a two-factor split-plot experimental design with varieties × seedlings per hill. The number of seedlings per hill was divided into split plots, and the varieties were the main plots. The treatments were randomly arranged among the main plots of four different cultivars and four seedling patterns (T1 = 1 seedling (12.93 × 12.93 cm), T2 = 3 seedlings (22.33 × 22.33 cm), T3 = 6 seedlings (31.67 × 31.67 cm), and T4 = 9 seedlings (38.80 × 38.80 cm). The experiment consisted of sixteen treatment combinations. The basic number of transplanted seedlings of all varieties was 60 seedlings m^−2^. In October 2019, the 0–10 cm layer of the experimental field’s soil was excavated, air-dried, and sieved. In April 2020, a plot was made where the surface soil had been excavated. Wooden boards were cut to the plots’ lengths (or widths) (given in Table 10); they were 3 cm thick and 15 cm wide. The boards were glued together to form plot cells. The sieved soil was then backfilled into the plots to the height of 10 cm. The main interval is kept at a distance of 50 cm, which is convenient for field management and investigation, and the split intervals are side by side without interval. A 5 cm depth gap was left on one board of each plot to facilitate irrigation and drainage. Each plot was drained and irrigated separately.

Rice was managed by conventional seedlings and field management. The row spacing, plant spacing, and the number of seedlings per hill are shown in Table 10. The recommended dose of N-P-K fertilizer 180 kg N, 180 kg K_2_O, and 90 kg P_2_O_5_ per hectare was applied, respectively. The nitrogen, phosphorus, and potassium fertilizers applied in the experiment were urea, superphosphate, and potassium chloride, respectively. Nitrogen and potassium fertilizers were applied in three split doses, i.e., 50% basal dose, 30% tillering stage dose, and 20% panicle initiation stage dose. All phosphorus fertilizers were applied in the basal fertilizer dose. The basal fertilizer was applied two days before transplanting.

### 4.3. Soil Properties

Soil samples were randomly collected from each treatment to determine basic soil properties. The soil pH was measured with a desktop pH meter. The contents of soil organic matter, available nitrogen, total phosphorus, and total potassium were measured by potassium dichromate volumetric method-external heating method, alkaline hydrolysis diffusion method, NaOH melting-molybdenum antimony anti-colorimetry method, and NaOH melting-flame photometry, respectively [41]. Soil organic matter (SOM) content was estimated by multiplying soil organic carbon by 1.72 [42,43].

### 4.4. Yield and Yield Components

At maturity, plants from three hills were sampled. The number of effective panicles was investigated in four yield components: number of effective panicles, spikelets per panicle, percentage of filled grains, and 1000-grain weight. The rice in each plot was harvested, and then the rice yield per unit area was calculated according to the weight of the harvested rice.

### 4.5. Biomass and Total Nitrogen

The three-hill plant samples were split into three parts: leaves, stems, and panicle and placed in an oven at 105 °C for 30 min, then dried to a constant weight in an 85 °C oven and weighed. The dried stem, leaf, and ear samples were crushed, passed through a 100-mesh sieve, and boiled with H_2_SO_4_-H_2_O_2_, and the total nitrogen content was determined by a continuous flow chemical analyzer.

### 4.6. Data Processing

Rice panicle biomass was defined as grain biomass, denoted as GB. Leaf biomass per plant was expressed as LB. The total biomass of leaf sheaths and stems was taken as stem biomass, denoted by SB. The sum of the biomass of stems, leaves, and ears of a single plant was taken as the aboveground biomass, which was expressed as AGB. The reproductive leaf ratio was defined as the ratio of the total biomass of the reproductive organs of a single plant to the biomass of the leaf organs. Biomass allocation was calculated using the formula of ODM/TDM, where ODM was the total dry matter of panicle, leaf, and stem with leaf sheath (referred to as the stem). TDM was the total dry matter of aboveground organs.

The allometric equation can be used to describe the mechanism of allometric growth [44,45]:*Y* = *K*_i_M^b^
where *Y* is biological characteristic (organ size); M is individual size; *K*_i_ is a species-specific constant, and b is the allometric index. When b = 1, there is isokinetic growth between individual biological characteristics and individual size; when b ≠ 1, there is allometric growth [12].

Standardized Major Axis Tests were conducted in the ‘SMATR’s package in R (4.1.2) [46]. There are four main allometric relationships between different species and organs [47,48]. Type A: significantly different slopes; Type B: significantly different intercepts, but with a common fit axis (the same slope); Type C: the same slopes, the same intercepts, but drift along the common fit axis; Type D: drift B and C occurs at the same time, i.e., the intercepts are different, and the fitted axes are not the same (the slopes are different).

The N accumulation, N use efficiency (NUE), and N harvest index (NHI) were computed as follows [17]:
Nitrogen Use Efficiency CalculationFormulaAbbreviations and UnitsN accumulation by grainN content of grain × grain weightGN, kg hm^−2^N accumulation by stemN content of stem × stem weightSN, kg hm^−2^N accumulation by leafN content of leaf × leaf weightLN, kg hm^−2^Total N accumulationGN + LN + SNTNA, kg hm^−2^N dry matter production efficiencytotal above-ground biomass/TNANDMPE, kg kg^−1^N grain production efficiencyrice grain yield/TNANGPE, kg kg^−1^N harvest indexGN/(GN + LN + SN)NHI

### 4.7. Statistical Analysis

Data among varieties and treatments were analyzed by ANOVA, and the mean comparisons between treatments were made using Duncan’s multiple range test. Statistical significance was taken at *p* < 0.05 and *p* < 0.01. All data analyses were performed using SPSS 26.0 software (SPSS Inc., Chicago, IL, USA).

## 5. Conclusions

Among the four rice varieties, the hybrid rice ‘Yexiangyou2’ showed a greater yield advantage. Planting one seedling per hill and maintaining a narrow row spacing can obtain higher effective panicle number and grain biomass distribution, and at the same time, improve nitrogen dry matter production efficiency and nitrogen harvest index, and increase yield. The number of seedlings per hill and row spacing affect growth and yield mainly by affecting the growth relationship between various organs of the rice plant. Under the condition of the same basic seedling number, when one seedling was transplanted per hill and planted with narrow row spacing, the allometric relationship between leaves and other organs increased, and the ratio of grain to leaf increased significantly. Therefore, transplanting one seedling per hill (narrow row spacing) is more conducive to shaping the high-yield phenotype of rice.

## Figures and Tables

**Figure 1 plants-11-02508-f001:**
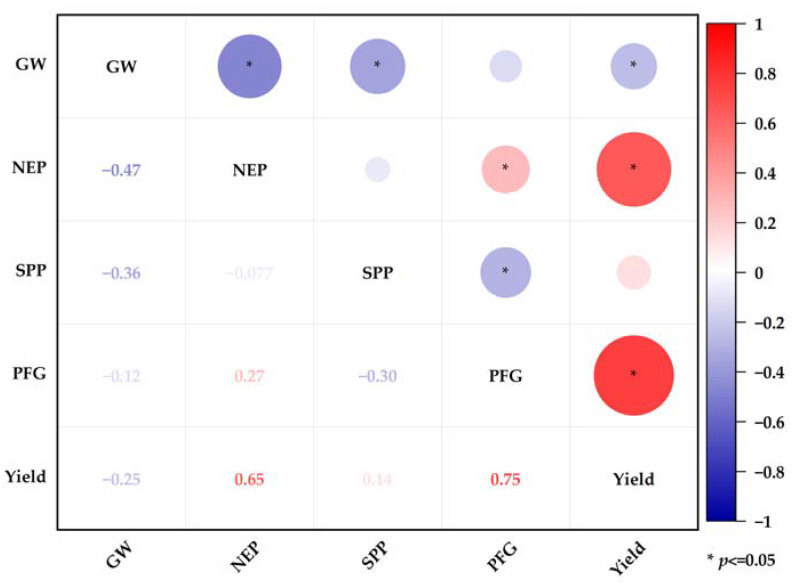
Heatmap of correlations between yield and yield components. Note: GW: 1000-grain weight; NEP: number of effective panicles; SPP: spikelets per panicle; PFG: percentage of filled grains; * indicates significance at 0.05.

**Figure 2 plants-11-02508-f002:**
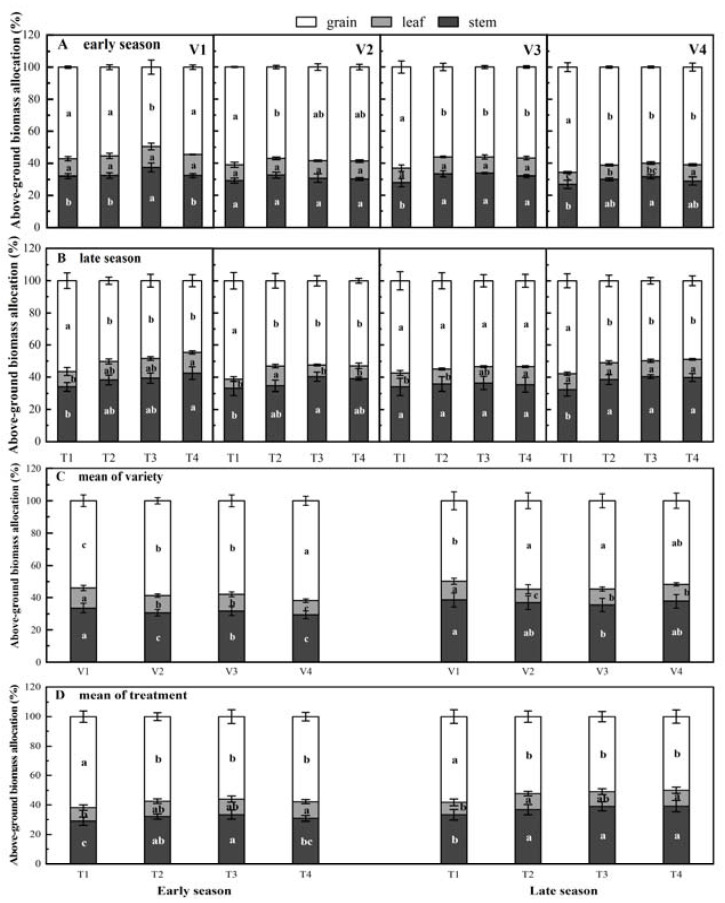
Effects of different treatments on above-ground biomass allocation of early and late season rice. Note: T1, 1 seeding per hill + row spacing 12.93 cm; T2, 3 seedings per hill + row spacing 22.33 cm; T3, 6 seedings per hill + row spacing 31.67 cm; T4, 9 seedings per hill + row spacing 38.80 cm; V1, V2, V3, and V4 represent ‘Guiyu’, ‘Zhenguiai’, ‘Zhuangxiangyou5’, and ‘Yexiangyou2’; different lowercase letters within subgraphs indicate significant differences between treatments (*p* < 0.05); The longer the error bars, the greater the variability of the sample data in the group.

**Figure 3 plants-11-02508-f003:**
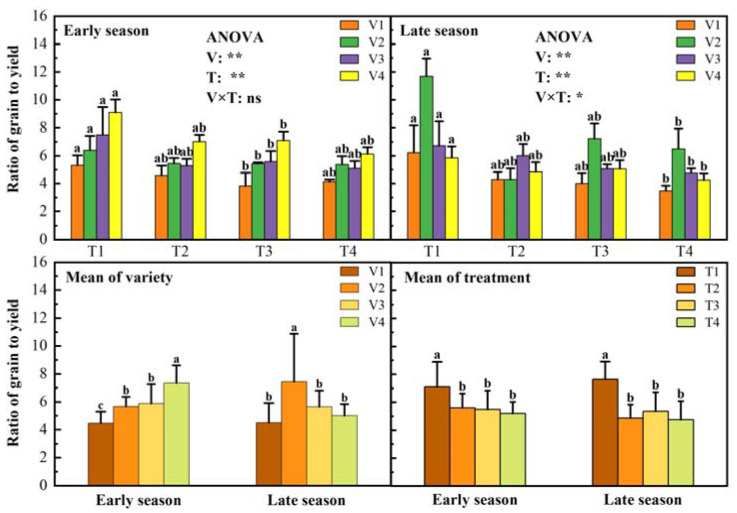
Effects of different treatments on grain to leaf ratio of rice. Note: T1, 1 seeding per hill + row spacing 12.93 cm; T2, 3 seedings per hill + row spacing 22.33 cm; T3, 6 seedings per hill + row spacing 31.67 cm; T4, 9 seedings per hill + row spacing 38.80 cm; different lowercase letters within subgraphs indicate significant differences between treatments (*p* < 0.05); The longer the error bars, the greater the variability of the sample data in the group. ** indicates significance at 0.01, * indicates significance at 0.05, ns indicates no significance.

**Figure 4 plants-11-02508-f004:**
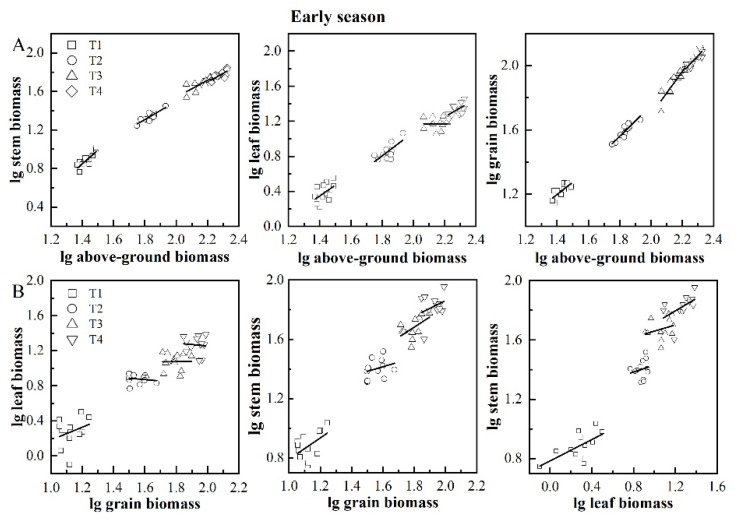
The allometric relationship between total aboveground biomass and organ biomass (**A**) and between organ and organ biomass (**B**) in early season rice. Note: T1, 1 seeding per hill + row spacing 12.93 cm; T2, 3 seedings per hill + row spacing 22.33 cm; T3, 6 seedings per hill + row spacing 31.67 cm; T4, 9 seedings per hill + row spacing 38.80 cm.

**Figure 5 plants-11-02508-f005:**
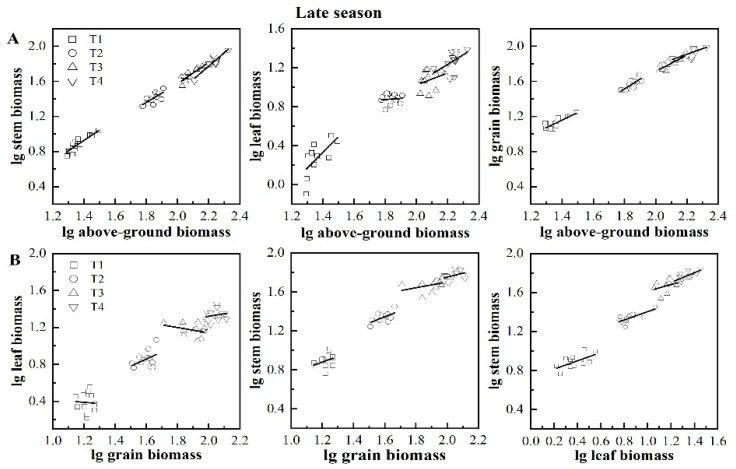
The allometric relationship between total aboveground biomass and organ biomass (**A**) and between organ and organ biomass (**B**) in late season rice. Note: T1, 1 seeding per hill + row spacing 12.93 cm; T2, 3 seedings per hill + row spacing 22.33 cm; T3, 6 seedings per hill + row spacing 31.67 cm; T4, 9 seedings per hill + row spacing 38.80 cm.

**Table 1 plants-11-02508-t001:** Effects of seedling numbers per hill and row spacing on yield of different rice varieties (t ha^−1^).

Treatment	V1	V2	V3	V4	Treatment Mean	F-Value		
						Variety (V)	Treatment (T)	V × T
Early season								
T1	6.88 ± 0.70 a	7.44 ± 0.60 a	6.34 ± 2.46 a	8.11 ± 1.46 a	7.19 ± 1.46 a	8.88 **	2.90 *	1.02
T2	6.78 ± 1.01 a	5.66 ± 0.39 ab	6.50 ± 1.05 a	8.35 ± 0.36 a	6.82 ± 1.21 ab			
T3	5.29 ± 0.49 a	6.67 ± 0.80 b	6.98 ± 1.14 a	7.98 ± 0.42 a	6.73 ± 1.20 ab			
T4	5.23 ± 1.00 a	5.73 ± 0.43 b	5.75 ± 0.76 a	7.37 ± 0.52 a	6.02 ± 1.04 b			
Variety mean	6.05 B	6.38 B	6.39 B	7.95 A				
Late season								
T1	5.49 ± 0.82 a	6.14 ± 1.08 a	6.18 ± 1.29 a	7.23 ± 0.69 a	6.26 ± 1.07 a	4.70 **	10.08 **	0.69
T2	5.20 ± 0.75 a	5.59 ± 1.04 a	6.40 ± 1.42 a	5.76 ± 0.86 b	5.74 ± 1.00 ab			
T3	4.36 ± 0.73 ab	5.16 ± 0.81 a	4.70 ± 0.46 a	5.90 ± 0.64 b	5.03 ± 0.83 bc			
T4	3.77 ± 0.55 b	5.00 ± 0.18 a	4.47 ± 0.61 a	4.94 ± 0.30 b	4.55 ± 0.64 c			
Variety mean	4.71 B	5.47 AB	5.44 AB	5.96 A				

Note: V1, V2, V3, and V4 represent ‘Guiyu’, ‘Zhenguiai’, ‘Zhuangxiangyou5’, and ‘Yexiangyou2’, respectively; ** indicates significance at 0.01, * indicates significance at 0.05. Within a column, different lowercase letters indicate significant differences between treatments, On the same row, different capital letters indicate significant differences between breeds (*p* < 0.05).

**Table 2 plants-11-02508-t002:** Effects of different seedling numbers per hill and row spacing on rice yield components.

Treatment	GW (g)	NEP (10^4^ ha^−1^)	SPP	PFG (%)
Early season				
T1	22.41 ± 2.08 a	256.67 ± 37.98 a	153.98 ± 22.75 a	0.82 ± 0.11 a
T2	22.66 ± 1.81 a	237.78 ± 21.52 ab	159.63 ± 22.20 a	0.80 ± 0.10 a
T3	21.83 ± 1.75 a	250.55 ± 32.25 ab	159.3 ± 19.48 a	0.78 ± 0.09 a
T4	21.31 ± 1.72 a	227.78 ± 28.45 b	169.37 ± 14.73 a	0.74 ± 0.10 a
*p*-vaule	0.291	0.109	0.311	0.233
Late season				
T1	22.46 ± 1.81 a	233.33 ± 44.59 ab	191.11 ± 24.48 a	0.64 ± 0.07 a
T2	22.45 ± 1.56 a	252.22 ± 34.91 a	165.94 ± 16.09 b	0.62 ± 0.08 a
T3	22.38 ± 2.01 a	221.39 ± 33.77 b	165.45 ± 15.40 b	0.62 ± 0.07 a
T4	22.24 ± 2.05 a	208.71 ± 25.68 b	168.32 ± 11.62 b	0.59 ± 0.08 a
*p*-vaule	0.911	0.029	0.002	0.443

Note: Within a column, different lowercase letters indicate significant differences between treatments (*p* < 0.05). GW: 1000-grain weight; NEP: number of effective panicles; SPP: spikelets per panicle; PFG: percentage of filled grains.

**Table 3 plants-11-02508-t003:** ANOVA of above-ground biomass allocation among variety and treatments.

ANOVA	Early Season			Late Season		
	Stem	Leaf	Grain	Stem	Leaf	Grain
Variety (V)	**	**	**	ns	**	*
Treatment (T)	**	**	**	**	**	**
V × T	ns	ns	ns	ns	**	ns

Note: ** indicates significance at 0.01, * indicates significance at 0.05, ns indicates no significance.

**Table 4 plants-11-02508-t004:** Test of allometric growth between the aboveground biomass and organ biomass in early and late season rice.

Treatment	X	Y	Allometric Growth Index			95% CI	Isokinetic Test		Type
			R^2^	*p*	A (Slope)		r	*p*	
Early season									
T1	AGB	SB	0.708	0.001	1.694	1.167~2.456	0.714	0.009	AR
		LB	0.249	0.099	2.482	1.393~4.425	0.768	0.004	AR
		GB	0.6	0.003	0.986	0.640~1.518	−0.023	0.944	IR
T2	AGB	SB	0.761	0	1.082	0.772~1.517	0.159	0.621	IR
		LB	0.511	0.009	1.88	1.170~3.022	0.694	0.012	AR
		GB	0.829	0	1.047	0.785~1.397	0.111	0.731	IR
T3	AGB	SB	0.56	0.005	1.168	0.744~1.835	0.229	0.474	IR
		LB	0.998	0	−1.254	−2.418~−0.650	0.223	0.487	IR
		GB	0.795	0	1.499	1.095~2.053	0.676	0	AR
T4	AGB	SB	0.61	0.003	1.153	0.753~1.767	0.223	0.485	IR
		LB	0.348	0.044	1.713	0.996~2.946	0.573	0.051	AR
		GB	0.767	0	1.182	0.847~1.652	0.329	0.296	IR
Late season									
T1	AGB	SB	0.742	0	1.378	0.970~1.957	0.54	0.07	IR
		LB	0.416	0.024	2.492	1.488~4.172	0.807	0.002	AR
		GB	0.738	0	0.982	0.690~1.399	−0.035	0.915	IR
T2	AGB	SB	0.558	0.005	1.421	0.904~2.235	0.475	0.119	IR
		LB	0.01	0.755	1.166	0.606~2.241	0.153	0.635	IR
		GB	0.71	0.001	1.329	0.917~1.926	0.472	0.121	IR
T3	AGB	SB	0.781	0	1.381	0.999~1.910	0.575	0.051	IR
		LB	0.11	0.293	1.838	0.985~3.429	0.566	0.055	IR
		GB	0.718	0	1.096	0.760~1.580	0.17	0.597	IR
T4	AGB	SB	0.819	0	1.693	1.260~2.275	0.792	0.002	AR
		LB	0.243	0.104	2.004	1.122~3.579	0.654	0.021	AR
		GB	0.419	0.023	0.984	0.588~1.645	−0.022	0.946	IR

Note: T1, 1 seeding per hill +row spacing 12.93 cm; T2, 3 seedings per hill +row spacing 22.33 cm; T3, 6 seedings per hill + row spacing 31.67 cm; T4, 9 seedings per hill +row spacing 38.80 cm; AR: allometric relationship; IR: isokinetic growth relationship; AGB: above-ground biomass; GB: grain biomass; LB: leaf biomass; SB: stem biomass.

**Table 5 plants-11-02508-t005:** Test of allometric growth between organ biomass in early and late season rice.

Treatment	X	Y	Allometric Growth Index			95% CI	Isokinetic Test		Type
			R^2^	*p*	A (Slope)		r	*p*	
Early Season									
T1	GB	LB	0.003	0.864	−2.518	−4.851~−1.307	0.728	0.007	AR
		SB	0.123	0.264	1.719	0.925~3.194	0.519	0.084	IR
	LB	SB	0.39	0.03	0.682	0.403~1.154	−0.448	0.144	IR
T2	GB	LB	0.183	0.165	1.795	0.985~3.272	0.565	0.055	AR
		SB	0.386	0.031	1.033	0.610~1.751	0.041	0.898	IR
	LB	SB	0.545	0.006	0.575	0.364~0.911	−0.653	0.021	AR
T3	GB	LB	0.118	0.274	−0.836	−1.556~−0.450	−0.188	0.559	IR
		SB	0.145	0.221	0.779	0.422~1.438	−0.263	0.409	IR
	LB	SB	0.127	0.255	0.932	0.502~1.728	−0.076	0.815	IR
T4	GB	LB	0.035	0.562	1.449	0.759~2.765	0.36	0.25	IR
		SB	0.167	0.187	0.975	0.532~1.787	−0.027	0.924	IR
	LB	SB	0.553	0.006	0.673	0.427~1.061	−0.52	0.083	IR
Late season									
T1	GB	LB	0.085	0.359	2.537	1.349~4.769	0.746	0.005	AR
		SB	0.262	0.089	1.402	0.790~2.488	0.372	0.233	IR
	LB	SB	0.424	0.022	0.553	0.331~0.923	−0.638	0.026	AR
T2	GB	LB	0.032	0.58	−0.877	−1.675~−0.459	−0.133	0.681	IR
		SB	0.086	0.354	1.069	0.569~2.010	0.07	0.83	IR
	LB	SB	0.035	0.559	1.219	0.639~2.362	0.199	0.535	IR
T3	GB	LB	0.000	0.964	1.677	0.870~3.233	0.476	0.118	IR
		SB	0.297	0.067	1.26	0.719~2.208	0.268	0.399	IR
	LB	SB	0.076	0.385	0.751	0.397~1.416	−0.289	0.363	IR
T4	GB	LB	0.006	0.807	−2.037	−3.920~−1.059	0.613	0.034	AR
		SB	0.093	0.335	1.721	0.918~3.228	0.514	0.088	IR
	LB	SB	0.244	0.103	0.845	0.473~1.509	−0.199	0.552	IR

Note: T1, 1 seeding per hill + row spacing 12.93 cm; T2, 3 seedings per hill + row spacing 22.33 cm; T3, 6 seedings per hill + row spacing 31.67 cm; T4, 9 seedings per hill + row spacing 38.80 cm; AR: allometric relationship; IR: isokinetic growth relationship; GB: grain biomass; LB: leaf biomass; SB: stem biomass.

**Table 6 plants-11-02508-t006:** Common slope test and allometric relationship and drift type of above-ground organ biomass in different treatments of rice.

Season	X	Y	Shift A Test		Shift B Test		Shift C Test		Shift Type
			LR	*p*	Wald	*p*	Wald	*p*	
Early season	AGB	SB	3.730	0.292	13.550	0.004	331.60	0.000	D
		LB	2.543	0.468	9.653	0.022	141.00	0.000	D
		GB	3.876	0.275	9.855	0.020	447.70	0.000	D
	GB	LB	6.234	0.101	3.919	0.270	126.90	0.000	C
		SB	3.519	0.318	9.164	0.027	192.7	0.000	D
	LB	SB	1.625	0.654	15.600	0.001	311.8	0.000	D
Late season	AGB	SB	1.336	0.721	11.590	0.009	330.80	0.000	D
		LB	3.402	0.334	12.330	0.006	111.80	0.000	D
		GB	1.778	0.620	3.596	0.309	318.60	0.000	C
	GB	LB	5.915	0.116	6.400	0.094	108.80	0.000	C
		SB	3.519	0.318	9.164	0.027	154	0.000	D
	LB	SB	3.829	0.281	10.080	0.018	191	0.000	D

Note: C: coaxial drift; D: intercept drift and coaxial drift; LR: likelihood ratio; Wald: Wald test.

**Table 7 plants-11-02508-t007:** Effects of different seedling numbers per hill and row spacing on nitrogen accumulation in aboveground organs of rice.

Variety	Treatment	SN (kg hm^−^^2^)		LN (kg hm^−2^)		GN (kg hm^−2^)		TNA (kg hm^−2^)	
		Early Season	Late Season	Early Season	Late Season	Early Season	Late Season	Early Season	Late Season
V1	T1	63.64 ± 15.42 a	45.22 ± 4.59 a	26.32 ± 2.32 ab	19.41 ± 5.03 b	161.24 ± 14.02 a	131.25 ± 10.73 a	251.20 ± 30.36 a	195.88 ± 8.53 ab
	T2	65.35 ± 16.18 a	57.78 ± 10.48 a	25.05 ± 7.24 ab	28.65 ± 1.92 a	174.57 ± 18.71 a	141.21 ± 19.12 a	264.97 ± 40.76 a	227.64 ± 29.88 a
	T3	63.50 ± 10.24 a	44.33 ± 8.11 a	33.13 ± 3.90 a	24.32 ± 0.96 ab	144.10 ± 24.75 a	108.11 ± 5.99 b	240.73 ± 20.87 a	176.76 ± 14.05 b
	T4	68.10 ± 5.11 a	46.56 ± 4.46 a	22.60 ± 5.98 b	29.15 ± 3.18 a	169.93 ± 24.64 a	108.59 ± 5.45 b	260.64 ± 33.66 a	184.30 ± 4.78 b
V2	T1	59.60 ± 7.05 a	32.80 ± 9.11 b	28.55 ± 6.80 a	10.83 ± 4.68 a	248.43 ± 18.02 a	154.52 ± 21.10 a	336.59 ± 30.57 a	198.15 ± 32.08 a
	T2	57.46 ± 9.08 a	39.14 ± 1.49 ab	28.24 ± 1.68 a	25.86 ± 2.33 b	176.05 ± 13.59 b	143.03 ± 26.90 a	261.75 ± 9.95 b	208.03 ± 25.98 a
	T3	66.24 ± 14.51 a	48.15 ± 9.07 a	34.45 ± 1.99 a	15.05 ± 1.17 b	188.31 ± 14.45 b	127.20 ± 20.09 a	289.01 ± 28.25 b	190.41 ± 23.93 a
	T4	49.52 ± 7.42 a	44.38 ± 4.99 ab	27.07 ± 3.29 a	17.91 ± 6.35 b	179.38 ± 11.78 b	125.62 ± 7.52 a	255.97 ± 15.20 b	187.91 ± 17.84 a
V3	T1	61.54 ± 10.87 ab	46.46 ± 5.60 a	30.75 ± 7.01 a	14.40 ± 2.21 b	233.03 ± 24.78 a	171.80 ± 10.28 a	325.31 ± 11.71 ab	232.66 ± 14.63 a
	T2	55.57 ± 4.05 b	45.72 ± 2.33 a	25.89 ± 0.65 a	17.21 ± 2.31 ab	209.42 ± 21.61 a	171.34 ± 31.02 a	290.88 ± 24.35 b	234.27 ± 31.36 a
	T3	71.47 ± 7.16 a	38.30 ± 4.25 ab	33.28 ± 3.08 a	17.87 ± 0.97 ab	245.27 ± 12.42 a	161.55 ± 21.13 a	350.02 ± 16.04 a	217.72 ± 26.02 a
	T4	63.86 ± 11.01 ab	31.29 ± 7.38 b	26.54 ± 2.70 a	18.99 ± 2.92 a	207.41 ± 23.19 a	152.22 ± 30.29 a	297.82 ± 31.37 b	202.50 ± 40.46 a
V4	T1	50.28 ± 12.85 a	42.84 ± 15.45 a	18.14 ± 3.38 b	29.54 ± 5.63 a	219.28 ± 19.22 a	216.54 ± 36.16 a	287.70 ± 34.16 a	288.92 ± 48.52 a
	T2	46.58 ± 0.94 a	61.81 ± 7.06 a	20.17 ± 3.53 ab	28.46 ± 2.10 a	216.38 ± 5.68 ab	188.33 ± 35.46 ab	283.13 ± 7.45 a	278.60 ± 40.14 a
	T3	54.17 ± 8.85 a	55.34 ± 3.90 a	22.99 ± 6.25 ab	24.77 ± 1.46 a	215.27 ± 24.53 ab	178.95 ± 16.47 ab	292.43 ± 39.24 a	259.06 ± 21.59 a
	T4	46.52 ± 2.10 a	43.44 ± 7.84 a	26.47 ± 1.31 a	26.90 ± 6.69 a	179.42 ± 22.46 b	152.94 ± 9.20 b	252.42 ± 21.63 a	223.27 ± 22.24 a
Mean	T1	58.77 ± 11.50 ab	41.83 ± 9.98 b	25.94 ± 6.72 b	18.54 ± 8.32 c	215.49 ± 38.21 a	168.53 ± 37.72 a	300.20 ± 42.49 a	228.91 ± 47.00 ab
	T2	56.24 ± 10.69 b	51.11 ± 10.99 a	24.84 ± 4.67 b	25.05 ± 5.20 a	194.10 ± 24.09 b	160.98 ± 32.09 ab	275.18 ± 24.50 bc	237.14 ± 38.52 a
	T3	66.34 ± 15.49 a	46.53 ± 8.64 ab	30.96 ± 5.98 a	20.50 ± 4.46 bc	198.24 ± 42.37 b	143.95 ± 32.49 bc	293.05 ± 46.77 ab	210.99 ± 37.76 bc
	T4	57.00 ± 11.37 ab	41.42 ± 8.24 b	25.67 ± 3.69 b	23.24 ± 6.69 ab	184.04 ± 23.24 b	134.84 ± 24.10 c	266.71 ± 29.55 c	199.49 ± 26.57 c
Variety (V)		**	**	**	**	**	**	**	**
Treatment (T)		ns	**	**	**	**	**	*	**
V × T		ns	*	ns	*	**	ns	*	ns

Note: V1, V2, V3, V4 represent ‘Guiyu9’, ‘Zhenguiai’, ‘Zhuangxiangyou5’, and ‘Yexiangyou2’, respectively; ** and * indicate significance at 0.01 and 0.05 levels, respectively. ns: no significant. Within a column, different lowercase letters indicate significant differences between treatments. SN: N accumulation by stem; LN: N accumulation by leaf; GN: N accumulation by grain; TNA: total N accumulation.

**Table 8 plants-11-02508-t008:** Effects of different seedling numbers per hill and row spacing on nitrogen use efficiency of rice.

Variety	Treatment	N Dry Matter Production Efficiency (kg kg^−1^)		N Grain Production Efficiency (kg kg^−1^)		Nitrogen Harvest Index (%)	
		Early Season	Late Season	Early Season	Late Season	Early Season	Late Season
V1	T1	68.38 ± 1.00 a	65.58 ± 2.15 ab	27.45 ± 1.55 a	27.97 ± 3.51 a	0.64 ± 0.03 a	0.67 ± 0.04 a
	T2	57.01 ± 1.71 b	60.47 ± 2.16 b	25.61 ± 1.64 b	22.94 ± 2.93 ab	0.66 ± 0.04 a	0.62 ± 0.00 b
	T3	52.73 ± 0.75 c	66.07 ± 4.64 a	21.97 ± 0.58 c	24.89 ± 5.28 ab	0.59 ± 0.06 a	0.61 ± 0.02 b
	T4	50.78 ± 3.95 c	62.49 ± 0.34 ab	19.95 ± 1.80 c	20.41 ± 2.49 b	0.65 ± 0.02 a	0.59 ± 0.02 b
V2	T1	46.18 ± 1.42 a	67.24 ± 3.74 a	22.11 ± 0.26 a	30.98 ± 1.62 a	0.74 ± 0.02 a	0.78 ± 0.04 a
	T2	45.72 ± 1.29 a	62.68 ± 3.10 a	21.61 ± 0.95 a	26.73 ± 1.75 b	0.67 ± 0.04 b	0.68 ± 0.04 b
	T3	45.92 ± 1.54 a	62.97 ± 4.87 a	23.08 ± 2.01 a	27.04 ± 1.36 b	0.65 ± 0.02 b	0.67 ± 0.04 b
	T4	45.25 ± 0.82 a	61.24 ± 3.37 a	22.41 ± 1.77 a	26.77 ± 2.72 b	0.70 ± 0.03 ab	0.67 ± 0.02 b
V3	T1	50.27 ± 1.92 a	58.60 ± 3.89 ab	16.22 ± 6.56 a	26.65 ± 6.13 a	0.76 ± 0.05 a	0.74 ± 0.02 a
	T2	48.36 ± 3.60 ab	59.89 ± 1.91 a	22.34 ± 3.19 a	27.14 ± 2.53 a	0.72 ± 0.02 ab	0.73 ± 0.04 a
	T3	44.90 ± 1.97 bc	51.05 ± 5.39 b	19.54 ± 3.92 a	21.88 ± 4.18 a	0.68 ± 0.03 b	0.74 ± 0.01 a
	T4	42.60 ± 2.10 c	51.30 ± 4.54 b	19.30 ± 1.22 a	22.50 ± 4.10 a	0.70 ± 0.02 b	0.75 ± 0.01 a
V4	T1	54.81 ± 1.04 a	55.80 ± 1.36 a	28.11 ± 3.04 a	25.26 ± 2.05 a	0.76 ± 0.03 a	0.75 ± 0.05 a
	T2	49.71 ± 1.05 a	53.51 ± 6.02 a	29.51 ± 1.96 a	20.80 ± 3.21 a	0.76 ± 0.01 a	0.67 ± 0.04 b
	T3	51.05 ± 4.88 a	55.24 ± 2.37 a	27.52 ± 2.84 a	22.76 ± 1.52 a	0.74 ± 0.02 ab	0.69 ± 0.01 ab
	T4	53.79 ± 3.70 a	56.36 ± 0.76 a	29.25 ± 1.50 a	22.23 ± 1.79 a	0.71 ± 0.03 b	0.69 ± 0.03 ab
Mean	T1	54.91 ± 8.81 a	61.81 ± 5.57 a	23.47 ± 5.92 a	27.71 ± 3.90 a	0.73 ± 0.06 a	0.73 ± 0.06 a
	T2	50.20 ± 4.74 b	59.14 ± 4.75 ab	24.77 ± 3.72 a	24.40 ± 3.58 b	0.70 ± 0.05 ab	0.68 ± 0.05 b
	T3	48.65 ± 4.19 bc	58.83 ± 7.31 ab	23.03 ± 3.77 a	24.14 ± 3.65 b	0.67 ± 0.06 c	0.68 ± 0.05 b
	T4	48.11 ± 5.24 c	57.85 ± 5.22 b	22.73 ± 4.33 a	22.98 ± 3.47 b	0.69 ± 0.03 bc	0.68 ± 0.06 b
Variety (V)		**	**	**	**	**	**
Treatment (T)		**	ns	ns	**	**	**
V × T		**	ns	ns	ns	ns	*

Note: V1, V2, V3, V4 represent ‘Guiyu9’, ‘Zhenguiai’, ‘Zhuangxiangyou5’, and ‘Yexiangyou2’, respectively; ** and * indicate significance at 0.01 and 0.05 levels, respectively. ns: no significant. Within a column, different lowercase letters indicate significant differences between treatments.

**Table 9 plants-11-02508-t009:** Differences in nitrogen accumulation and nitrogen use efficiency among four rice varieties.

Variety	SN (kg hm^−2^)	LN (kg hm^−2^)	GN (kg hm^−2^)	TNA (kg hm^−2^)	NDMPE (kg kg^−1^)	NGPE (kg kg^−1^)	NHI (%)
Early season							
V1	65.15 ± 10.88 a	26.77 ± 6.03 a	162.46 ± 21.65 c	254.38 ± 29.15 c	57.23 ± 7.38 a	23.75 ± 3.33 b	0.64 ± 0.04 d
V2	58.21 ± 10.54 ab	29.58 ± 4.54 a	198.04 ± 33.18 b	285.83 ± 38.49 b	45.77 ± 1.16 c	22.30 ± 1.34 b	0.69 ± 0.04 c
V3	65.60 ± 14.53 a	29.12 ± 4.71 a	223.78 ± 24.52 a	316.01 ± 30.98 a	44.36 ± 3.43 c	19.35 ± 4.22 c	0.72 ± 0.04 b
V4	49.38 ± 7.49 b	21.94 ± 4.74 b	207.59 ± 23.77 b	278.92 ± 29.23 b	52.34 ± 3.43 b	28.60 ± 2.23 a	0.74 ± 0.03 a
Late season							
V1	48.47 ± 8.46 a	25.38 ± 4.91 a	122.29 ± 18.02 c	196.15 ± 25.04 c	63.65 ± 3.37 a	24.05 ± 4.28 b	0.62 ± 0.03 c
V2	41.12 ± 8.45 b	17.41 ± 6.74 b	137.59 ± 21.24 c	196.12 ± 23.22 c	63.53 ± 4.02 a	27.88 ± 2.49 a	0.70 ± 0.06 b
V3	40.44 ± 7.84 b	17.12 ± 2.59 b	164.23 ± 22.65 b	221.79 ± 28.62 b	55.21 ± 5.51 b	24.54 ± 4.51 b	0.74 ± 0.02 a
V4	50.86 ± 11.71 a	27.42 ± 4.31 a	184.19 ± 33.09 a	262.46 ± 39.75 a	55.23 ± 3.05 b	22.76 ± 2.54 b	0.70 ± 0.04 b

Note: V1, V2, V3, V4 represent ‘Guiyu9’, ‘Zhenguiai’, ‘Zhuangxiangyou5’, and ‘Yexiangyou2’, respectively; Within a column, different lowercase letters indicate significant differences between treatments; SN: N accumulation by stem; LN: N accumulation by leaf; GN: N accumulation by grain; TNA: total N accumulation.

**Table 10 plants-11-02508-t010:** Test plot area and plant-row spacing settings.

Item	Seedlings per Hill	Basic Number of Seedlings (m^−2^)	Total Number of Seedlings per Plot	Plot Area (m^2^)	row Spacing (cm)
T1	1	60	225	3.75	12.93 × 12.93
T2	3	60	243	4.05	22.33 × 22.33
T3	6	60	216	3.60	31.67 × 31.67
T4	9	60	225	3.75	38.80 × 38.80

## Data Availability

Not applicable.

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
