# Peer review of "Allometric Characteristics of Rice Seedlings under Different Transplanted Hills and Row Spacing: Impacts on Nitrogen Use Efficiency and Yield"

_plants, 2022, doi:10.3390/plants11192508_

Round 1

Reviewer 1 Report

Dear Authors,

thank you for the opportunity to meet the manuscript entitled: "Allometric characteristics of seedlings in different transplanted hills under the same basic seedlings and the effects on nitrogen use efficiency and yield".

In relation to food self-sufficiency, it is necessary to investigate the possibilities of increasing plant yields. One possibility is research in the field of planting method and its effect on selected rice traits.

The submitted manuscript is extensive, it covers a sufficient number of rice traits, although it is currently desirable to supplement the basic traits with various physiological indices. In addition, I have major reservations and many comments regarding the preparation of the manuscript.

First, the manuscript is not submitted by the Plants journal template. Please follow the journal requirements.

I also recommend linguistic and stylistic editing of the text.

The composition of the manuscript is at a poor level. Tables and figures should be as close as possible to the text where they were first mentioned. In the MM chapter, individual information is scattered without logical continuity, which has a distracting effect.

Tables and figures are not prepared precisely. In some, the data is incorrectly stated, in some the data is missing, etc. This needs to be improved.

Abstract:

It is necessary to shorten the abstract. The background is unnecessarily extensive; I recommend only briefly indicating the main topic of the research.

The same information is found twice between L 13-16.

Introduction:

I also recommend reducing this chapter. The purpose of the Introduction chapter is to briefly and clearly define the importance of the study and to present the current state of the researched area. In this manuscript, several informations are unnecessarily repeated, which reduces its quality.

In addition, I recommend adding information about the novelty of this study at the end of the chapter.

Discussion:

This chapter is again extensive, but much of it duplicates the results chapter. I recommend paying close attention to only the most important findings of the experiment and comparing them with other works. I believe it would increase the quality of the manuscript.

Conclusions:

I recommend adding some information (see attachment)

Materials and methods:

This chapter needs to be given extra attention. A lot of information is inaccurate or completely missing.

Detailed comments are provided in the attachment.

Author Response

Dear Authors,

thank you for the opportunity to meet the manuscript entitled: "Allometric characteristics of seedlings in different transplanted hills under the same basic seedlings and the effects on nitrogen use efficiency and yield".

In relation to food self-sufficiency, it is necessary to investigate the possibilities of increasing plant yields. One possibility is research in the field of planting method and its effect on selected rice traits.

The submitted manuscript is extensive, it covers a sufficient number of rice traits, although it is currently desirable to supplement the basic traits with various physiological indices. In addition, I have major reservations and many comments regarding the preparation of the manuscript.

First, the manuscript is not submitted by the Plants journal template. Please follow the journal requirements.

Response: Thanks for your suggestion. We have changed the template in the revise version.

I also recommend linguistic and stylistic editing of the text.

Response. Thanks for your suggestion. We have revised the manuscript on native English speaker with a company named edit bar the certificate link is given below.

The composition of the manuscript is at a poor level. Tables and figures should be as close as possible to the text where they were first mentioned. In the MM chapter, individual information is scattered without logical continuity, which has a distracting effect.

Response: Thanks for your suggestion. We moved the table and approached the text describing it.

Tables and figures are not prepared precisely. In some, the data is incorrectly stated, in some the data is missing, etc. This needs to be improved.

Response: Thanks for your suggestion. We checked the data in the tables and charts, corrected the errors and added the data about the varieties.

Abstract:

It is necessary to shorten the abstract. The background is unnecessarily extensive; I recommend only briefly indicating the main topic of the research.

The same information is found twice between L 13-16.

Response: Thanks for your suggestion. The abstract has reduced.

Introduction:

I also recommend reducing this chapter. The purpose of the Introduction chapter is to briefly and clearly define the importance of the study and to present the current state of the researched area. In this manuscript, several informations are unnecessarily repeated, which reduces its quality.

In addition, I recommend adding information about the novelty of this study at the end of the chapter.

Response: Thanks for your suggestion. We shorten the introduction as according to your suggestion.

Discussion:

This chapter is again extensive, but much of it duplicates the results chapter. I recommend paying close attention to only the most important findings of the experiment and comparing them with other works. I believe it would increase the quality of the manuscript.

Response: Thank you. We followed your valuable suggestion in the revised version.

Conclusions:

I recommend adding some information (see attachment)

Response: Thank you for your kind suggestion and information.

Materials and methods:

This chapter needs to be given extra attention. A lot of information is inaccurate or completely missing.

Detailed comments are provided in the attachment.

Response: Thanks for your suggestion. We already changed the template in the revise version.

Detailed comments:

1.Table 1 shows only grain yield, not yield components. So I recommend mentioning Table 2, but it does not show the overall influence of the factor, as the authors mention it.

Response: Thanks for your suggestion. We have referred to Table 2 in the manuscript and added a correlation analysis of yield and yield components.

2.Please use consistent terminology throughout the manuscript.

Response: Thanks for your suggestion. We have changed "per mound" to "per hill"

  1. Edit based on journal requirements (t ha-1)

Response: Thanks for your suggestion. We have revised as requested in the manuscript.

  1. I'm not sure if the table shows the P-values. Please check!

Response: Thanks for your suggestion. We have changed "P-vaule" to "F-vaule" in Table 1.

  1. I assume that ANOVA was used to indicate the overall effect of the factor.

Response: Thanks for your suggestion. We have made changes.

  1. The order of the figures is incorrect. This is the first figure, so why is it labeled Fig.3?

Response: Thanks for your suggestion. We checked the order of the figure and edited the correct order.

  1. In the figure, this is marked as "reproductive tissues". However, I recommend using the term "grain".

Response: Thanks for your suggestion. We have changed "reproductive biomass" to "grain biomass" in the manuscript.

  1. I recommend inserting an explanation of abbreviations.

Response: Thanks for your suggestion. We have inserted notes for abbreviations (RB, LB, SN, etc.).

  1. This was not part of the research. Authors should avoid assumptions.

Response: Thanks for your suggestion. We avoided hypothetical results and inserted relevant references.

  1. Since different varieties of rice were part of the research, I recommend including the most important findings about them in the Conclusions. Which one is the most suitable in relation to the examined signs and parameters?

Response: Thanks for your suggestion. We have added an analysis of varieties differences to the manuscript.

  1. What does "the early and late seasons" mean? It is necessary to specify the planting and harvesting dates.

Response: Thanks for your suggestion. We mention the detailed months of the early and late seasons in the manuscript.

  1. What are the recommended doses? Were they applied based on the results of the analyses?

Response: Thanks for your suggestion. We put the details of fertilization in 4.1 into 4.2 for explanation

  1. I recommend authors to use the Statistical analysis chapter separately. The current form is inappropriate. Moreover, the method by which the relationships were analyzed is missing.

Response: Thanks for your suggestion. We have divided statistical analysis and data processing into two chapters

Reviewer 2 Report

The manuscript entitled “Allometric characteristics of seedlings in different transplanted hills under the same basic seedlings and the effects on nitrogen use efficiency and yield” invested the growth pattern of seedlings under different transplanted condition and the effects of different seedlings per hill on yield formation and nutrient use efficiency of rice. My suggestion is major revision.

1. Please revise the title and make it more intuitive and easier understandable for reader

2. Abstract is too long.

3. Line 8-9: according to the environment?

4. It is suggested to provide some exact values from your results in the Abstract.

5. Line19-20: the maximum number of effective panicles were recorded in 1 seedling treatment? It seems to be inconsistent with your results.

6. It suggested to place the Materials and methods behind the Introduction and front the Results.

7. Line413-415: So it was not just seedlings number differed from each treatment and there was also transplanting density treatment in your study? It is suggested to revise your treatment description and objective throughout the manuscript as well as title.

8. There are grammar mistakes in the manuscript. Please carefully check and corrected them.

Author Response

The manuscript entitled “Allometric characteristics of seedlings in different transplanted hills under the same basic seedlings and the effects on nitrogen use efficiency and yield” invested the growth pattern of seedlings under different transplanted condition and the effects of different seedlings per hill on yield formation and nutrient use efficiency of rice. My suggestion is major revision.

  1. Please revise the title and make it more intuitive and easier understandable for reader

Response: Thanks for your suggestion. We have changed the title to be more convenient for the readers.

  1. Abstract is too long.

Response: Thank you for your suggestion. We shorten the abstract.

  1. Line 8-9: according to the environment?

Response: Thanks for your suggestion. We already changed the template in the revise version.

  1. It is suggested to provide some exact values from your results in the Abstract.

Response: Thank you for your suggestion. We followed your valuable suggestion.

  1. Line19-20: the maximum number of effective panicles were recorded in 1 seedling treatment? It seems to be inconsistent with your results.

Response: Thanks for your suggestion. We revised it version.

  1. It suggested to place the Materials and methods behind the Introduction and front the Results.

Response: Thank you. We followed the journal template.

  1. Line413-415: So it was not just seedlings number differed from each treatment and there was also transplanting density treatment in your study? It is suggested to revise your treatment description and objective throughout the manuscript as well as title.

Response: We have revised treatment description and objective according to your valuable suggestion.

  1. There are grammar mistakes in the manuscript. Please carefully check and corrected them.

Response: Thanks for your suggestion, we followed your valuable suggestion.

Reviewer 3 Report

Please find enclosed document.

Author Response

The paper entitled „Allometric characteristic of seedlings in different transplanted hills under the same

basic seedlings and the effects on nitrogen use efficiency and yield” focuses on the important problem

related to methods to increase crop yields and nitrogen use efficiency.

This manuscript is well written and in my opinion it only needs minor corrections, as below:

  1. Lines 13-16: the sentence “Therefore, a field experiment….” is repeated.

Response: Thanks for your suggestion. We already changed the template in the revise version.

  1. Line 30-31: allometric growth, yield, number of seedlings…

Response: Thanks for your suggestion. We already changed the template in the revise version.

  1. Line 19: nitrogen

Response: Alright. Thank you

  1. Line 117-120: in Table 2 there was find the significant differences between T1 and T4 in SPP .in the late season too

Response: Thank you for your suggestion. We have corrected this error in the revised version.

  1. Line 129: Table 2.

Response: Thank you for your suggestion. We have corrected this error in the revised version.

  1. Line 142: its true but for late season

Response: Thank you for your suggestion. We have corrected this error in the revised version.

  1. Line 159: Figure 3. The description – please explain what mean the letters and the the error bars

Response: Thank you for your suggestion. We explained it in  the revised version.

  1. Line 156: Figure 4.

Response: Thank you for your suggestion. We have corrected this error in the revised version.

  1. Line 173: please add the description of SN, LN, GN and TNA under the Table 3

Response: Thanks for your suggestion. We added the description of the table in the revised version.

  1. Line 196: NUEb – please explain in the text

Response: Thanks for your suggestion. It is explained in the revise version.

  1. Line 203: NUEg – as above

Response: Thank you. Noted.

  1. Line 244: please add the SB and its explanation under Table 5

Response: Thanks for your suggestion. We already changed the template in the revise version.

  1. Lines 417 and 429: Table 8 instead of Table 1

Response: Thank you. Replaced in the revised version.

  1. Line 429-430: please explain what kind of N, P, K fertilizers were applied in the experiment

Response: Thanks for your suggestion. We explained it in the revised version.

  1. Lines 449-450: please change the font

Response: Thanks for your suggestion. We already changed the template in the revised version.

  1. References, please check this part and insert a comma after the year

Response: Thanks for your suggestion. We already changed the template in the revised version.

Round 2

Reviewer 1 Report

The authors have expended sufficient effort in revising the article. I believe that editing the manuscript based on my comments helped to improve its quality.

Despite the above, I recommend paying extra attention to the formal editing of the text and tables.

Chapter 4.7 needs to be corrected. The authors reported that ANOVA and two-way ANOVA were used. This is the same analysis, so it is unnecessary to mention "ANOVA" twice. Additionally, Duncan's test, not ANOVA, was used to analyze differences between treatments and varieties.

In the Conclusions, it is mentioned that hybrid rice showed a greater yield advantage. Given that two hybrid rices were examined in the experiment, which one did the authors have in mind? It is necessary to be precise.

Reviewer 2 Report

The Authors have replied to the numerous questions of the reviewers in a sufficient manner, to my opinion. Now, the work seems more clear than before and it can be considered for publication.